# Agreement Between Consumer and Research-Grade Physical Activity Monitors in a Public Health Intervention for Adolescent Latinas

**DOI:** 10.3390/ijerph22111663

**Published:** 2025-11-02

**Authors:** Jacob Carson, David Wing, Job G. Godino, Michael Higgins, Britta Larsen

**Affiliations:** 1Herbert Wertheim School of Public Health & Human Longevity Science, University of California, La Jolla, San Diego, CA 92093, USA; dwing@health.ucsd.edu (D.W.); jobg@fhcsd.org (J.G.G.); mdhiggins@health.ucsd.edu (M.H.); blarsen@health.ucsd.edu (B.L.); 2Exercise and Physical Activity Resource Center, University of California, San Diego, CA 92093, USA; 3Laura Rodriguez Research Institute, Family Health Centers of San Diego, San Diego, CA 92101, USA

**Keywords:** accelerometry, activity monitoring, consumer wearables, MVPA

## Abstract

Consumer wearables are increasingly used in physical activity (PA) interventions, but their validity as a measurement tool among low PA groups, like adolescent girls, is unclear. We assessed the minute- and day-level agreement between PA measures among adolescent Latinas from an intervention. Participants wore a Fitbit Inspire HR and an ActiGraph GT3X+ for overlapping epochs. ActiGraph data were classified using two different cut points and aligned with Fitbit data to produce 1,149,169 matched minutes of wear across 137 adolescent girls (M = 15.73 yrs). Confusion matrices were calculated for pairwise comparisons to determine minute-level Moderate-Vigorous PA (MVPA) classification. Data were aggregated to 1007 days for Bland–Altman analyses. ActiGraph cut points showed moderate agreement for minute-level MVPA classification (Balanced Accuracy = 0.71, AC1 = 0.98), while Fitbit showed fair agreement (Balanced Accuracy = 0.50, AC1 = 0.95–0.97) largely driven by non-MVPA observations. The Freedson cut point overestimated daily MVPA relative to Treuth by 14.7 min/day and Fitbit by 14.2 min/day in Bland–Altman space. The daily Treuth and Fitbit comparison did not significantly differ. Findings suggest systematic differences between cut points that warrant further consideration. Fitbit showed moderate agreement with ActiGraph, but heteroscedasticity and the epoch of aggregation significantly impacted agreement. Understanding device differences has implications for promoting/researching public health among adolescents.

## 1. Introduction

The measurement of free-living (i.e., non-laboratory) physical activity (PA) is important for the evaluation of health behavior interventions, scientific rigor, and increasingly personalized health recommendations. Understanding how various technologies estimate PA intensity among different age, gender, and racial/ethnic groups is important for rigorous research, establishing construct validity, and providing accurate PA guidance [1]. Research-grade hip-worn accelerometer devices, such as the ActiGraph GT3X+, have been long held as the gold standard for measuring free-living PA. However, these devices have a number of limitations. ActiGraphs are particularly high burden for participants, are high cost, and are often worn only for short periods of time (usually 5–7 days) which are assumed to be representative of longitudinal activity. Additionally, quantification of free-living PA duration and intensity using ActiGraph and similar accelerometer-based devices relies on applying cut points to raw data. While cut points for healthy adults have been well studied, there are no gold standard cut points for other populations, including children/adolescents [2,3]. Identifying acceptable, accurate methods for measuring free-living activity among the most inactive populations, such as Latina adolescents [4], is imperative for PA promotion research.

Consumer wearables, such as those made by Fitbit, have been increasingly used in research studies. While they are typically used as intervention tools, they also have some potential advantages over ActiGraphs in measuring PA. They are lower burden for study participants, have higher compliance, and can reduce researcher burden, while still providing interpretable granular data [5,6]. Additionally, Fitbits and other consumer wearables can classify PA through multi-sensor processes, using both heart rate and acceleration. This theoretically allows for the more accurate capture of intensity for exercises such as weight lifting or cycling that may not be well characterized by ActiGraph accelerometer cut points alone, and allows for individualized classification of activity rather than applying standard cut points to all individuals [7,8,9].

Despite the potential benefits of using Fitbits, existing literature shows mixed results in the agreement between Fitbit devices and other objective measures of PA in children/adolescents. Compared to laboratory research grade measures (electrocardiogram and indirect calorimetry), Fitbit devices showed 85.4% accuracy in classifying Moderate-Vigorous Physical Activity (MVPA) among boys and girls ages 9–11 [9]. Also in a lab setting, Fitbit Charge HR devices showed validity in measuring both sedentary behavior and MVPA in a sample of 43 young children with superior intensity classification due to heart rate measurement compared to a wrist-worn ActiGraph device [10]. However, a recent study published by Schmidt et al. compared Fitbit Flex 2 and ActiGraph GT9X+ in 123 elementary school children and suggested only fair to good agreement [11]. Spearman correlations of free-living MVPA minutes in that sample varied substantially depending on the ActiGraph cut points used (Evenson vs. Romanzini), with moderate (rs = 0.54 to 0.55) or vigorous (rs = 0.29 to 0.48) PA. Other free-living comparisons in adolescents have suggested moderate agreement, with Fitbit tending to overestimate steps [12]. In addition to mixed findings, there is limited research across different demographic groups. Identifying the optimal tools for the measurement of free-living PA, especially for inactive groups like adolescent females, promotes high quality, equitable, and actionable research.

Currently, ActiGraph-measured PA intensity depends on the cut-point classifications used, and there is no consensus on the appropriate cut points for adolescent girls [3]. Understanding how different devices classify clinically meaningful outcomes, such as MVPA, or whether PA guidelines are being met, can provide researchers with important context for the practical implications of their work [13]. Accurately assessing PA is a particularly important goal for designing successful interventions among inactive groups like adolescent girls [4]. To our knowledge, this is the among the first studies to explore agreement between the Fitbit and ActiGraph devices with different cut points among Latina adolescent girls or adolescent girls generally.

The aim for this manuscript is to identify the agreement between ActiGraph and Fitbit measures of free-living MVPA in a sample of Latina teens in Southern California (ages 13–18). Additionally, the ActiGraph comparison was performed with two different cut points given the lack of a standard procedure in this population and to provide insights into the impacts of cut-point choices consistent with previous research.

## 2. Materials and Methods

### 2.1. Study Design and Participants

This study used baseline and 6-month data from the Chicas Fuertes Randomized Controlled Trial, a mobile technology-based PA intervention for Latina adolescents based in Southern California (R01NR017876). Chicas Fuertes included 160 Latina Teens (ages 13–18) living in Southern California who reported participating in less than 150 min of MVPA per week and being in the first three stages of change regarding PA at enrollment (Pre-Contemplation, Contemplation, or Preparation) [14]. The study received IRB approval, and all participants provided informed assent/consent. More detailed eligibility criteria and recruitment for the parent study can be found in the protocol paper [15].

The present analyses assessed measurement agreement between concurrently gathered estimates of PA intensity using a matched cross-sectional design where the unit of observation was a single minute or a valid day between both measures (defined below) for any given participant.

### 2.2. Measures

#### 2.2.1. Demographics

Demographic information was ascertained via a self-reported questionnaire at baseline and included age, race, parents’ education, income, and the number of children in the home. The Stages of Change for Physical Activity (SCPA) questionnaire was used to stage-match participants [14].

#### 2.2.2. Measures of Physical Activity

PA was measured using two different validated measures. A hip-worn ActiGraph GT3X+ Accelerometer captured PA for seven days at baseline, then again at six months, while the wrist-worn Fitbit Inspire HR was worn continuously throughout the course of the study, including the weeks when the ActiGraph was worn.

The ActiGraph GT3X+ is a triaxial accelerometer commonly worn on the hip that measures movement in order to identify activity intensity. It has been validated against heart rate telemetry [16] and total energy expenditure in children [17,18]. Accelerometers were delivered via mail with detailed instructions to wear the accelerometer on their left-side hip for an entire waking day (approximately 12 h) for 7 consecutive days. Valid wear time was defined as wearing the device for at least 600 min per day for five days or a total of at least 3000 min over four days, with wear time identified using the algorithm developed by Choi et al. [19].

The Fitbit Inspire HR uses a MEMS triaxial accelerometer and optical heart rate tracker that combine to determine PA intensity through a proprietary algorithm. Four potential levels of activity intensity are used: sedentary, light, moderate, and very active [20]. This device has been validated in adults and older adults. Further, the full product range of Fitbit devices have been validated in hundreds of papers across all age groups, including adolescents [21,22,23]. Fitbit Inspire HR devices were worn continuously throughout the entire 12-month follow-up period for each participant, with regular text message reminders to sync the device to ensure complete data capture.

### 2.3. Data Processing and Alignment

For minute-level data, a valid observation was defined as any minute that had both ActiGraph and Fitbit data. For the analysis using day-level data, a valid day for ActiGraph was defined as having >600 min of wear time. For Fitbit, a valid day was defined as having >600 min of heart-rate data (in alignment with ActiGraph and previous research in children) OR >6000 steps in a day as to not erroneously exclude Fitbit observations with valid PA intensity data [24]. This approach for defining a valid day of Fitbit wear by also using step data was to catch participants with inconsistent heart rate data that can occur from improper wearing of the device (e.g., over a shirt) and is consistent with approaches in previous studies from our team to exclude step counts that could be erroneous [25,26,27,28]. An observation for the day-level analysis was only valid if a subject had valid data on both devices for any given day (i.e., met both of the criteria for ActiGraph and Fitbit described above).

For the ActiGraph GT3X+, two different cut points were applied given the lack of a standard in adolescent girls: Freedson (Adult) and Treuth (Children) [29,30]. Freedson was chosen given its ubiquity in the literature and previous validation in different comparisons [3]. Treuth was used given that it was developed specifically in girls ages 13–14, and thus is better aligned to our current sample [30]. Cut points were applied to the “Axis 1” (vertical axis) data captured in device-specific counts per minute (CPM) to create two distinct PA measures. For Freedson the applied cut points were Sedentary: 0–99 CPM; Light: 100–1951 CPM; Moderate: 1952–5724 CPM; Vigorous: 5725–9498 CPM; and Very Vigorous: 9499–∞ CPM [29]. For Treuth the applied cut points were Sedentary: 0–99 CPM; Light: 100–2999 CPM; Moderate: 3000–5200 CPM; and Vigorous: 5201–∞ CPM [30]. Given that the Treuth algorithm and Fitbit devices do not have a “Very Vigorous” category, this was collapsed into Vigorous so that all measures were comparable.

Complete Fitbit data for all participants over the entire study period was downloaded from Fitabase, an online data management platform for Fitbit devices (Small Steps Lab, LLC, San Diego, CA, USA). Given that ActiGraphs were only worn for discrete periods (7–14 days dependent on if valid data were gathered) while Fitbit data were available continuously throughout the entire study period, Fitbit data were matched to the existing ActiGraph data. Minute-level data were matched by participant ID and a complete timestamp containing date and time. Day-level data were created by summing the minutes of MVPA for every valid day, matched by participant ID and date. For the minute-level and day-level data, observations that were missing (NA values) either ActiGraph or Fitbit intensity were excluded from the sample for all analysis.

### 2.4. Statistical Analyses

Basic descriptive analyses to capture frequencies and percentages were performed on race, parents’ education, income, and SCPA. Mean and standard deviations were calculated for age and number of children in the home. All statistical analyses were completed in R version 4.2.1 (23 June 2022) and RStudio version 2023.09.0+463. Bland–Altman analyses were performed using the *blandr* package and by hand to verify consistent outputs.

#### 2.4.1. Minute-Level Agreement (Objective PA)

Confusion matrices, a type of contingency table that compares “actual” vs. “predicted” values to determine agreement, were created for MVPA vs. not-MVPA for each pairwise comparison between the ActiGraph (two cut points) and Fitbit devices (2 × 2 tables). Analyses were restricted to MVPA vs. non-MVPA to align with the goals of the study and provide consistency with PA recommendations. Matrices are presented as heat maps to display the trends of PA and relative agreement. For each matrix, Accuracy, Balanced Accuracy, and Gwet’s AC1 were calculated. Gwet’s AC1, similar to Cohen’s Kappa, is an agreement statistic looking at the chance that the classification agreement observed could be by chance. Gwet’s AC1 was chosen over Fleiss’ Kappa given its relative stability for unbalanced data [31].

#### 2.4.2. Day-Level Agreement (Objective PA)

At the day-level, total minutes of Moderate and Vigorous activity were summed to calculate the total minutes of MVPA for matched days between ActiGraph and Fitbits. To aid in the interpretation of the results and explore the practical implications of the different measurements, the percentage of days categorized as meeting PA guidelines (>60 min of MVPA/day) was calculated for each device. Means and differences in MVPA were calculated and used to perform Bland–Altman analyses to identify agreement and bias between measures. Data were tested for normality and transformed when appropriate, in addition to performing stratified Bland–Altman analyses by activity level and comparing proportional differences. Given that interpretations across the different comparison methods were consistent with standard approaches, the simplest findings are presented here. In addition to agreement statistics and bias between measures presented in minutes of MVPA, Bland–Altman Plots are provided for all comparisons.

## 3. Results

### 3.1. Descriptive Statistics

Of the 160 participants in the parent study, 137 participants had at least one matched minute of PA data between the two wearable devices. Demographic data for these individuals are summarized in Table 1. The sample had a mean age of 15.7 (SD) years, predominantly identified as racially White (51.8%) and/or Other (43.8%) and second generation immigrants (70.1%). Most participants had parents with education levels less than a college graduate (71.5%) and annual incomes under $75,000 (75.8%). Nearly 83% of participants were in the “Preparation” stage of change for PA.

### 3.2. Minute-Level MVPA Classification

In total, complete data for 1,149,169 min were available for minute-level analyses across the two ActiGraph wear periods.

Agreement statistics for binary minute-level classification of MVPA are presented in Table 2. For the two ActiGraph cut points, there was classification accuracy of over 98%, a balanced accuracy of about 71%, and an AC1 of 0.98. Fitbit showed higher agreement with the Treuth cutpoint than Freedson across all metrics, with the balanced accuracy taking the largest dip across the three metrics within the ActiGraph comparison (around 50%). Figure 1 depicts the confusion matrices used to determine agreement between devices, with counts and percentages of agreement highlighted in green.

### 3.3. Day-Level Bland–Altman Analyses

For the day-level analyses, there were a total of 1007 matched days. The Freedson cut point resulted in the highest percentage of valid days being categorized as meeting PA guidelines (10.33%), followed by the Fitbit (6.45%), with the Treuth cut point resulting in the lowest percentage (2.48%).

With the exception of the Treuth vs. Fitbit comparison, all Bland–Altman comparisons showed statistically significant differences between the measures (Table 3). Given the non-normality of the data, log transformations, proportional difference plots, and subgroup analyses across the different levels of MVPA (Mean daily MVPA < 10 min, 10–30 min, and >30 min) were performed. Data presented here are non-transformed to show differences in minutes of MVPA.

#### 3.3.1. Treuth vs. Freedson Cut Points

The Bland–Altman analysis comparing daily minutes of MVPA between the Treuth and Freedson cut points reveals a mean difference of −14.7 min (SD: 13.7) with the limits of agreement ranging from −41.5 to 12.1, indicating that, on average, Freedson cut points yields higher MVPA levels than those recommended by Treuth. A visual inspection of the Bland–Altman plot depicts a down-sloping funnel shape, with Freedson overestimating compared to Treuth and increases in this overestimation at higher levels of MVPA (Figure 2). This is reinforced by the strong negative correlation (−0.75) between the difference and the mean and the high bias correction factor (0.73). Treuth and Freedson cut points exhibited a high correlation (0.90). Analyses on the log-transformed data and proportional differences plot were consistent with the results described above. Subgroup analyses showed significant overestimation in all three groups.

#### 3.3.2. Treuth vs. Fitbit

When comparing daily minutes of MVPA between the ActiGraph (Treuth) and Fitbit we found a non-significant mean difference of −0.5 min (SD: 25.9) with broad limits of agreement (−51.3 to 50.3). These results indicate that, on average, the MVPA is not different between Fitbit and ActiGraph using Treuth cut points. The Bland–Altman plots depict a pronounced funnel shape with a slight tendency to skew negative (particularly at higher levels of MVPA) suggesting larger disagreement at greater levels of PA (Figure 2). These visual trends are echoed in the fit statistics, which suggest a fair correlation between Treuth and Fitbit measures (r = 0.31, CCC = 0.28), with a small negative bias (Fitbit overestimation) when using the Treuth cut point (bias correction factor of 0.90). A logarithmic transformation of the data revealed an opposite direction of bias in the Treuth–Fitbit comparison, with Treuth significantly overestimating relative to Fitbit by 0.63 log units (SD: 1.83) for the daily comparison. Subgroup analyses by the amount of MVPA reported reveal overestimation by Treuth relative to Fitbit in low-activity segments (Mean MVPA < 10 min) of the sample (bias = 0.4 min, *p* = 0.02) and overestimation by Fitbit relative to Treuth in the high-activity segments (Mean MVPA > 30 min) of the sample (bias = 7.1 min, *p* = 0.07).

#### 3.3.3. Freedson vs. Fitbit

Between the ActiGraph (Freedson) and Fitbit, we found a mean difference of 14.2 min (SD: 29.35), with limits of agreement ranging from −43.3 to 71.7. These results indicate that, on average, the Freedson cut point reports significantly higher amounts of MVPA than the Fitbit. Once again, the Bland–Altman plot depicts a clear funnel shape without a pronounced slope and increasing dispersion/disagreement present at greater levels of PA. This corresponds with a fair correlation between measures (0.37) with a near-zero correlation between the difference and the mean (0.03), suggesting less heteroscedasticity despite the visualization of the data (Figure 2). This is underscored by substantial bias (BCF = 0.87), resulting in overall fair agreement between the two measures (CCC = 0.32). The log-transformed comparison results were consistent with the non-transformed findings. All log-transformed and subgroup analyses were consistent with the main group findings, reinforcing the presence of overestimation by Freedson relative to Fitbit.

## 4. Discussion

This study sought to assess the agreement between intensity classifications of PA using two cut points for ActiGraph devices and Fitbits in a sample of Latina adolescents. Overall Fitbit exhibited fair to moderate agreement in measuring MVPA with ActiGraph devices, but this was highly dependent on the level of aggregation, with substantial disagreement at the minute level (when assessing the confusion matrices) and a relative smoothing at the day level. Fitbit agreement was strongest with the Treuth cut point when aggregated to the day level, but agreement was still dependent on activity levels. Relative to the Treuth cut point, Fitbit classified 2.6 times as many days as meeting PA guidelines, while the Freedson cut point classified 4.2 times as many days as meeting PA guidelines. Practically, this indicates that researchers using PA guidelines as a primary outcome may be overestimating adolescent female MVPA or intervention effects if they use Fitbit, but more so if they use the Freedson cut point.

Binary classification for a given minute as MVPA between devices showed strong agreement when looking at accuracy and Gwet’s AC1 but less so for balanced accuracy, which is more sensitive to unbalanced data like those in this study. However, the graphical depiction of the confusion matrices (Figure 1) shows that this agreement is likely entirely driven by the “Not MVPA” agreement. At the day level, the ways that disagreement between these objective measures manifest becomes more complex. Freedson cut points estimated significantly higher daily MVPA compared to both the Treuth cut point and Fitbit, whereas there was no statistically significant difference in daily MVPA between Treuth cut points and Fitbit. Given the non-normality of the data, logarithmic transformations and subgroup analyses were performed to improve interpretability. While these analyses did not change the interpretation in the Freedson comparisons, log transformation suggested that Treuth significantly overestimated daily MVPA relative to Fitbit, likely driven by the lowest activity observations (<10 min MVPA per day). The minute-level confusion matrices (Figure 1) highlight that while the day-level aggregation results in no mean differences between Treuth and Fitbit, they are classifying different minutes as MVPA, implying a fundamental disagreement in what movements are considered MVPA. This may mean that while these devices disagree on what constitutes MVPA at the minute level (potentially due to the proprietary Fitbit algorithm or the differences in epochs of measurement), when aggregated to the day (where research findings may be more meaningful) those differences become less apparent, particularly for the more inactive participants. Minute level disagreement could also be due to the relatively small number of minutes that either device classified as MVPA given the largely inactive study sample.

The fact that Freedson cut points produced higher estimates of MVPA than Treuth is not surprising given the higher count threshold for MVPA in the Treuth cut points (1952 CPM vs. 3000 CPM, respectively), and our findings are consistent with a systematic review from Kim 2012 that highlighted the potential for Treuth to underestimate activity relative to Freedson across multiple studies [3]. The application of cut points is an important consideration for assessing the prevalence of meeting PA guidelines, as shown by previous research from Watson 2014 that compared cut points from nine different studies [32]. Published literature comparing a variety of cut points in a cohort of younger adolescent males and females suggests that Evenson and Freedson cut points outperform Treuth in intensity classification [33]. However, limited quality research in a laboratory setting has been performed to develop or adopt cut points that are specific to adolescent females. With the exception of Treuth, most cut points do not differentiate by sex, despite well documented differential metabolic equivalents by sex and age [34]. Our findings further suggest the need for consistency in approaches to selecting cut points for the sample population and more laboratory validation for different demographic groups. For the former, inconsistent cut-point application may have effects on scientific reproducibility and the ability for meta-analyses to accurately compare existing research. While a discrepancy between the Freedson and Treuth minutes was expected a priori given the lower CPM threshold for Moderate activity in Freedson, this difference was pronounced in a sample of free-living participants, emphasizing how crucial cut-point selection is for estimating activity in this context.

The comparison between devices provides novel insight into how Fitbits differ from and agree with the commonly accepted “gold standard” for measuring free-living PA. Our findings between ActiGraph and Fitbit are partially inconsistent with a previous systematic review by Feehan 2018, which has generally confirmed that Fitbit overestimates multiple metrics of PA in free-living settings (including steps and MVPA), specifically for time spent in higher activity levels [6]. In the current study, Fitbit did not significantly disagree with estimates for daily MVPA compared to Treuth cut points and may actually underestimate activity in the least active adolescent girls, however the minute-level analyses suggest substantial disagreement for the classification of MVPA. To better interpret these findings, the next section is focused on methodological differences between devices.

### 4.1. Methodological Differences: Actigraph vs. Fitbit

The different ways that Fitbits and ActiGraph devices measure PA could provide some insight on areas of disagreement, and the interpretation of “objective” measurements of free-living PA more generally. In both cases, devices attempt to estimate MVPA rather than measuring it directly. ‘Moderate’ and ‘Vigorous’ activity are classified as such based partially on heart rate; thus, Fitbit and other consumer wearables may have an advantage as they classify activities using formulas that account for heart rate and acceleration, while ActiGraph is solely acceleration based. This allows consumer wearables to account for individual differences in activity intensities, and to capture activities that may raise heart rate without producing hip accelerations (such as cycling and weightlifting)—activities that ActiGraph cut points do not capture [9]. Additionally, other contextual factors such as elevation gain or carrying additional weight cannot be accounted for using accelerometry alone, whereas Fitbit may be able to identify the increased difficulty using heart rate. However, heart rate is also influenced by other factors such as stress and temperature [35], which could cause the Fitbit to overestimate activity in those contexts relative to ActiGraph (measuring relative intensity vs. absolute movement). This, along with the limitations of the ActiGraph’s 60 s measurement epoch (as compared to potentially shorter Fitbit epochs) could be responsible for the difference in which minutes were classified as MVPA, wherein accelerometry captured activity while it was occurring and the Fitbit captured delayed heart rate increases.

While ActiGraph devices are typically considered the gold standard, they are reliant on population averages to establish cut-point thresholds that, as previously described, can have a significant impact on the amount of PA measured. While cut points do exist for specific demographics, such as adolescents, these are often based on research that used very small sample sizes which could produce large amounts of error given variability in factors such as step cadence and body size [3]. For example, Treuth cut points are based on 74 girls ages 13–14 from a single site, which may not be representative of other adolescent girls given rapid physiological changes during adolescence, therefore limiting generalizability of that approach [30,36]. Additionally, data aggregation epochs of 60 s, like those used by ActiGraph accelerometers, can miss shorter bursts of activity that could be captured (or overestimated) by the shorter aggregation intervals that can be used by Fitbit devices. This shorter aggregation is also subject to bias to noise or short movements, especially if the device is worn somewhere with more common spurious movements (such as the wrist).

While consumer wearables have considerable advantages, including better compliance and the ability to measure MVPA continuously over long time spans rather than relying on pre- post-measurements from research-grade accelerometry alone (especially in youth), there are limitations to using these devices for research. The proprietary nature of Fitbit devices and data limits the comparison across new products and full assessment of methodologies that could provide researchers and public health practitioners with confidence in the measurement. This proprietary nature limits replicability across studies and comparability with different devices at different time points. The limitations to both objective PA measures require researchers to carefully interpret the real-world meaning of their findings for participants and the population at large.

### 4.2. Strengths and Limitations

This study adds to our understanding of objective measurements of free-living PA in youth. Specifically, findings provide data for adolescent females in whom PA measurement agreement has been understudied. PA research that focuses on adolescent girls will benefit from the context this work provides on estimates of agreement and bias, especially if they are considering the use of Fitbits or other consumer measures of PA in their work. The analysis of multiple epochs, cut points, and comparison methodologies provides a robust comparison between measures. Additionally, the sample size in this study is significantly larger than previous research, which results in robust findings.

This study is not without its limitations. Given the use of free-living data in an age group without consensus on the ideal ActiGraph cut points, there is no “gold standard” or laboratory confirmed measurement by which to compare agreement, which limits the interpretability of the findings. Minute-level interpretations of agreement statistics in the literature are often inconsistent and therefore assigning terms such as “moderate” and “fair” to agreement between devices reflects a holistic interpretation of the analyses presented as opposed to standardized interpretations. This is especially true given that our sample was highly inactive, limiting the number of observations where agreement in MVPA classification could be compared. Given the imbalanced data, those agreement statistics should be interpreted with caution. Present analyses were limited to days/minutes where participants wore ActiGraph devices; this limits our ability to account for differences in PA patterns that may occur during ActiGraph wear. Additionally, our study population being from a single US city and demographic group could limit generalizability to other populations.

## 5. Conclusions

In summary, the findings of this study suggest that Fitbits provide moderate to strong agreement with ActiGraph in measuring MVPA in adolescent Latinas, but bias and heteroscedasticity underly measurement disagreements and the epoch of interest plays a role in agreement between devices. Additionally, the differences between Treuth and Freedson cut points highlight how cut-point decision points can substantially impact research findings and interpretation. These results contribute to a larger discussion on the sole reliance on accelerometry as the “gold standard” for measuring free-living PA. High-quality health promotion through PA interventions relies on the accurate measurement of PA, and this research adds to our understanding for a low-activity demographic. Analyzing performance differences across different activity contexts could elucidate when certain measures may be preferrable for unbiased activity estimates. Future research should focus on studying the specific activities, environmental contexts, and populations where Fitbit and ActiGraph devices differ to better understand agreement and measurement for approaches to health promotion.

## Figures and Tables

**Figure 1 ijerph-22-01663-f001:**
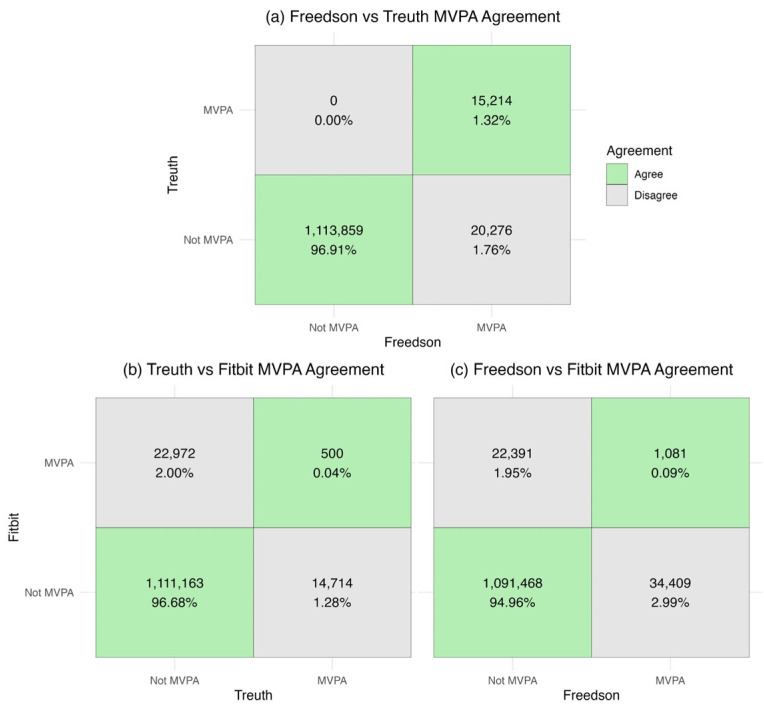
Minute-level binary MVPA agreement (**a**) Freedson vs. Treuth; (**b**) Treuth vs. Fitbit; (**c**) Freedson vs. Fitbit.

**Figure 2 ijerph-22-01663-f002:**
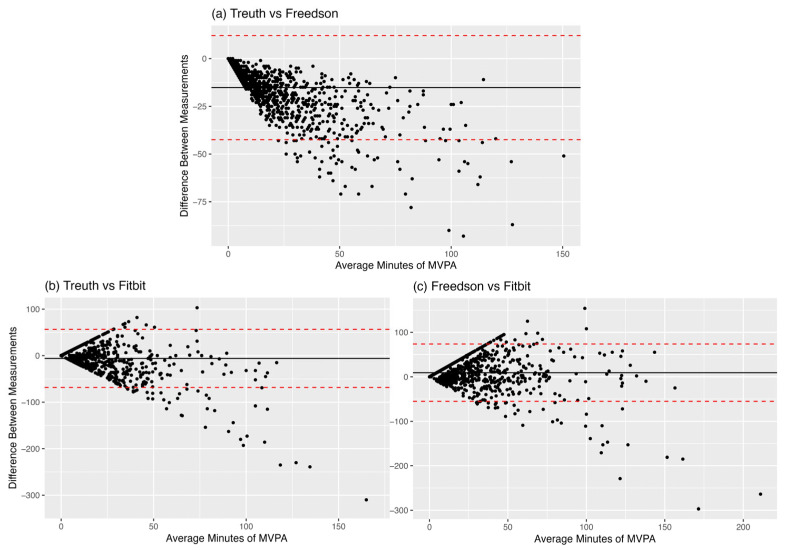
Day-level Bland–Altman Plots (**a**) Trueth vs. Freedson MVPA; (**b**) Trueth vs. Fitbit MVPA; (**c**) Freedson vs. Fitbit MVPA. The black line represents the mean difference (in minutes) between measures, while the red lines represent the limits of agreement.

**Table 1 ijerph-22-01663-t001:** Demographics (n = 137).

Characteristic	Category	n (%) or Mean (SD)
Age (years)		15.7 (1.6)
Race *	White	71 (51.8%)
	Black	3 (2.2%)
	Asian	4 (2.9%)
	American Indian or Alaskan Native	7 (5.1%)
	Native Hawaiian or Pacific Islander	0 (0%)
	Other	60 (43.8%)
Parent Education Level	Less than High School	41 (29.9%)
	High School Graduate or GED	32 (23.4%)
	Some College or Associate’s Degree	25 (18.2%)
	College graduate or Baccalaureate Degree	20 (14.6%)
	Master’s Degree	16 (11.7%)
	Professional/Vocational Degree	0 (0.00%)
	Doctoral Degree	3 (2.2%)
Household Income	Less than $11,999	25 (18.2%)
	$12,000 through $24,999	18 (13.1%)
	$35,000 through $49,999	40 (29.2%)
	$50,000 through $99,999	33 (24.1%)
	$100,000 or greater	21 (15.3%)
Generational Status	First Generation	10 (7.3%)
	Second Generation	96 (70.1%)
	Third Generation	31 (22.6%)
Number of Children in the Home		2.01 (1.18)
Physical Activity Stage of Change	Pre-Contemplation	4 (2.9%)
	Contemplation	19 (13.9%)
	Preparation	113 (82.5%)
	Missing	1 (0.7%)

* Participants were able to select multiple options for Race.

**Table 2 ijerph-22-01663-t002:** Agreement in binary minute-level classification of MVPA.

Agreement Metric	Freedson vs. Treuth	Treuth vs. Fitbit	Freedson vs. Fitbit
Accuracy	0.9824	0.9824	0.9567
Balanced Accuracy	0.7144	0.5043	0.5029
Gwet’s AC1	0.9816	0.9730	0.9547

**Table 3 ijerph-22-01663-t003:** Bland–Altman Comparisons.

Comparison	Bias	SD	LOA	Pearson	Mean-Difference Correlation	CCC	BCF
Treuth–Freedson	−14.72	13.67	(−41.51, 12.07)	0.90	−0.75	0.66	0.73
Treuth–Fitbit *	−0.50	25.93	(−51.33, 50.33)	0.31	−0.44	0.28	0.90
Freedson–Fitbit	14.22	29.35	(−43.30, 71.74)	0.37	−0.03	0.32	0.87

* Logarithmic adjustments changed the direction of bias, but subgroup analyses helped to contextualize those results. PAR: Physical Activity Recall Survey. LOA: Limits of Agreement. CCC: Correlation Concordance Coefficient. BCF: Bias Correction Factor.

## Data Availability

Data will be made available on request.

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
