# Peer review of "Agreement Between Consumer and Research-Grade Physical Activity Monitors in a Public Health Intervention for Adolescent Latinas"

_ijerph, 2025, doi:10.3390/ijerph22111663_

Round 1

Reviewer 1 Report

Comments and Suggestions for Authors

This review summarises the strengths of the manuscript, identifies gaps in methodology and results presentation, and suggests ways to improve the clarity and consistency of the study.

In the Introduction:

The statement (lines 54-56): “Additionally, Fitbits and other consumer wearables can classify physical activity through multi-sensor processes, using both heart 53 rate and acceleration. This may allow the accurate capture of intensity for exercises such as weight lifting or cycling that may not be well characterized by ActiGraph accelerometer cut points, and allows for individualized classification of activity rather than applying standard cut points to all individuals” requires greater caution and more robust references, as its validity in adolescents is questionable.

Lines 72-75 – The justification for focusing on Latina adolescents appears only at the end and is weak; it should be better substantiated earlier in the introduction.

Regarding Materials and Methods:

Lines 111-114 – There is confusion between data collected at baseline and at 6 months. This needs clarification, as line 92 states: “This study used baseline data from the Chicas Fuertes Randomized Controlled Trial...”

Lines 134-135 – The choice of >6000 steps or >600 minutes of HR for Fitbit could be supported with more detail from the literature that underpins this approach.

The alternation between “very vigorous” and “vigorous” (lines 151-152) may be confusing; ensure terminology is consistent throughout the manuscript.

Always use “physical activity” or “PA” consistently after the first mention.

In the Results:

“Trueth” (lines 216-217, Figure 2) is incorrect; the correct spelling is “Treuth.” This is important because it refers to a specific cut point; maintaining consistency is critical for interpretation.

Lines 224-227 and Table 3 – It is unclear whether the values presented are from the original data or log-transformed; this should be explicitly stated, as it affects statistical interpretation.

In the Discussion:

The percentage of days meeting guidelines (lines 285-286) lacks clinical/practical context; it would be helpful to express real-world implications.

Minute-by-minute disagreement versus daily agreement (lines 300-304) is well discussed, but it should be explicitly stated that differences may result from Fitbit algorithms and the ActiGraph 60-second epoch.

The limitations of the ActiGraph 60-second epoch (lines 356-368) should be reinforced as possible sources of deviation in minute-by-minute agreement.

Limitations due to the proprietary nature of Fitbit (lines 373-377) should be made explicit in terms of replicability and comparability with other devices.

Claims that Freedson outperforms Treuth (lines 310-316) and comparisons with Fitbit (lines 329-336) should be supported with specific references.

Highlighting that Treuth cut points were validated in a small sample of girls (n=74, 13-14 years, ref 27) emphasises the limitation of generalisability.

Finally, while overall agreement between Fitbit and ActiGraph is clear, terms such as “moderate to strong agreement” are vague, and the impact of differences between cut points could be quantified; it is recommended to explicitly state generalisation limitations and provide more concrete suggestions for future research.

Author Response

Thank you for your review, please see the attachment.

Reviewer 2 Report

Comments and Suggestions for Authors

Dear Authors,

The paper has an interesting topic that may spark interest among readers. However, it uses local data, a small number of participants, and requires more scientific soundness in the introduction and improvements in the methodological aspects used.

These aspects preclude publication in a high-impact international journal.

Author Response

(The authors gave the same response as above.)

Reviewer 3 Report

Comments and Suggestions for Authors

The introduction provides sufficient background and includes all valuable information for the research topic.

Among demographic questions, the authors mentioned marital status. It seems it has no relevance, as time is not used in the analysis. I recommend eliminating this parameter from the text.

The article title refers to a Public Health Intervention for Adolescent Latinas. According to the demographic data, it appears that there are multiple races, but the percentage of Latinas in the research sample is unclear. Are they the majority to justify the title?

Suppose the study population is limited to Latina adolescents in Southern California. In that case, may it restrict the external validity of the findings to other demographic groups or geographic locations, even though PA among adolescent girls is declining in many parts of the world?

Are these adolescents enrolled in a form of education?

The authors meticulously compare classification at both minute and day levels, revealing that while Fitbit devices demonstrate fair to moderate agreement with ActiGraph measures, the degree of agreement heavily depends on the aggregation level and the specific cut points applied. The difference in wear period (ActiGraph worn for 7-14 days versus Fitbit worn up to 1 year) may induce bias. Understandably, the ActiGraph wear is uncomfortable; however, the possibility of unmeasured confounders affecting PA patterns outside the ActiGraph period remains unaddressed.

This work provides empirical evidence on measurement agreement; stimulates critical discussion on the suitability of different devices and cut points for specific populations and research purposes, thereby contributing to the field of physical activity research. The authors acknowledge the absence of a proper “gold standard” for adolescent girls and the impact of cut point selection on outcomes.

The healthcare subject is rather implicit than explicit in this article. The final impression is that the comparison of the two devices is part of a healthcare program, as the title promises, but remains hidden from readers. 

Author Response

(The authors gave the same response as above.)
